# Adaptation of Partial Mutual Information from Mixed Embedding to Discrete-Valued Time Series

**DOI:** 10.3390/e24111505

**Published:** 2022-10-22

**Authors:** Maria Papapetrou, Elsa Siggiridou, Dimitris Kugiumtzis

**Affiliations:** Department of Electrical and Computer Engineering, Aristotle University of Thessaloniki, 54124 Thessaloniki, Greece

**Keywords:** Granger causality, conditional mutual information, mixed embedding, symbol sequences, discrete-valued time series, financial complex network

## Abstract

A causality analysis aims at estimating the interactions of the observed variables and subsequently the connectivity structure of the observed dynamical system or stochastic process. The partial mutual information from mixed embedding (PMIME) is found appropriate for the causality analysis of continuous-valued time series, even of high dimension, as it applies a dimension reduction by selecting the most relevant lag variables of all the observed variables to the response, using conditional mutual information (CMI). The presence of lag components of the driving variable in this vector implies a direct causal (driving-response) effect. In this study, the PMIME is appropriately adapted to discrete-valued multivariate time series, called the discrete PMIME (DPMIME). An appropriate estimation of the discrete probability distributions and CMI for discrete variables is implemented in the DPMIME. Further, the asymptotic distribution of the estimated CMI is derived, allowing for a parametric significance test for the CMI in the DPMIME, whereas for the PMIME, there is no parametric test for the CMI and the test is performed using resampling. Monte Carlo simulations are performed using different generating systems of discrete-valued time series. The simulation suggests that the parametric significance test for the CMI in the progressive algorithm of the DPMIME is compared favorably to the corresponding resampling significance test, and the accuracy of the DPMIME in the estimation of direct causality converges with the time-series length to the accuracy of the PMIME. Further, the DPMIME is used to investigate whether the global financial crisis has an effect on the causality network of the financial world market.

## 1. Introduction

A challenge in many domains of science and engineering is to study the causality of observed variables in the form of multivariate time series. Granger causality has been the key concept for this, where Granger causality from one variable to another suggests that the former improves the prediction ahead in time of the latter. Many methods have been developed based on the Granger causality idea to identify directional interactions among variables from their time series (see [1] for a recent comparative study of many Granger causality measures) and have been applied in various fields, such as economics [2], medical sciences [3], and earth sciences [4]. Of particular interest are measures of direct Granger causality that estimate the causal effect of the driving to the response variable that cannot be explained by the other observed variables. The estimated direct causal effects can then be used to form connections between the nodes being the observed variables in a causality network that estimates the connectivity (coupling) structure of the underlying system.

The studies on the Granger causality typically regard the continuous-valued time series, and often the number *K* of the time series is relatively high, and the underlying system is complex [5,6]. However, in some applications, the observations are discrete valued, e.g., the sign of the financial index return, the levels of precipitation, the counts of spikes in the electroencephalogram, and the counts of significant earthquake occurrences within successive time intervals. In this work, we propose an appropriate method to estimate the direct Granger causality on discrete-valued multivariate time series.

In the analysis of discrete-valued multivariate time series, termed symbol sequences when no order of discrete values is assumed, the causality effects are estimated typically in terms of the fitted model. Different approaches have been proposed for the form of the probability distribution models based on strong assumptions about the structure of the system, i.e., a multivariate Markov chain. One of the most known approaches giving a model of reduced form is the mixture transition distribution (MTD) [7], which restricts the initially large number of parameters, assuming that there is only an effect of each lag variable separately [8,9]. Recently, the MTD was adapted for the causality estimation in [10,11] and discussed in the review in [12]. Other approaches assume the Poisson distribution [13,14] and the negative binomial distribution [15,16]. In the category of autoregressive models are Pegram’s autoregressive models [17,18] and multivariate integer-valued autoregressive models (MINAR) [19,20]. Another proposed method, which, however, simplifies the problem to a linear one, is the so-called CUTE method [21]. Having obtained the model under given restrictions, one can then identify the causality of a driving variable to the response variable from the existence of lag terms of the driving variable in the model form.

Here, we follow a different approach and estimate the causality relationships directly using the information measures of mutual information (MI) and conditional MI (CMI). These measures have been employed to estimate causality and derive causality networks from continuous-valued time series [1,22,23]. For discrete-valued time series, MI and CMI have been used, e.g., for the estimation of the Markov chain order [24] and the estimation of autocorrelation in conjunction with Pegram’s autoregressive models [25]. They have also been used on discrete data derived from continuous-valued time series, either as ranks of components of embedding vectors [26,27] or as ordinal patterns [28,29,30]. However, we are not aware of any work on using information measures for a causality analysis of discrete-valued multivariate time series or symbol sequences (there is a reference to this in the supplementary material in [31]).

The framework of the proposed analysis is the estimation of the direct causality of a discrete driving variable *X* to a discrete response variable *Y* from the symbol sequences of *K* observed discrete variables, where *X* and *Y* are two of them. The direct causality implies the dependence of *Y* at one time step ahead, $Y_{t+1}$, on *X* at some lag τ≥0, $X_{t-\tau }$ that cannot be explained by any other variable at any lag. In the model setting, the direct causality is identified by the presence of the term $X_{t-\tau }$ in the model for $Y_{t+1}$. In the information theory setting, it is identified by the presence of significant information of $X_{t-\tau }$ for the response $Y_{t+1}$ that cannot be explained by other lag variables, which is quantified by the CMI of $X_{t-\tau }$ and $Y_{t+1}$ given the other lag variables. We develop this idea in a progressive algorithm that builds a set of the most informative lag variables for $Y_{t+1}$, called the discrete partial conditional mutual information from mixed embedding (DPMIME), based on a similar measure called the PMIME for continuous variables [32,33]. The presence of lag variables $X_{t-\tau }$ (for one or more different lags τ) in the derived set, the so-called mixed embedding vector, identifies the existence of the direct causality from *X* to *Y*, and the relative contribution of the lag variables of *X* in explaining $Y_{t+1}$ conditioned on the other components of the mixed embedding vector (regarding the other K−1 variables) quantifies the strength of this relationship. Further, we develop a parametric significance test for the CMI of the selected lag variable and $Y_{t+1}$ at each step of the DPMIME algorithm, which does not have an analogue in the PMIME regarding continuous variables.

In the evaluation of the DPMIME with Monte Carlo simulations, we compare the DPMIME to PMIME on discretized time series from continuous-valued systems and also discrete-valued time series generated by multivariate sparse Markov chains and MTD and MINAR systems, with a predefined coupling structure. We also compare the parametric significance test to the resampling significance test in the DPMIME. Further, we form the causality networks of five capital markets from the DPMIME, using the sign of the change in the respective daily indices, as well as other causality measures (computed on the values of the indices), and we compare the networks from each measure before and after the global financial crisis of 2008.

The structure of the paper is as follows. First, in Section 2, we present the proposed measure DPMIME along with the resampling and parametric test for the CMI. In Section 3, we assess the efficiency of the proposed DPMIME measure with a resampling and parametric test and compare the DPMIME to the PMIME in a simulation study. The results of the application regarding the global financial crisis are presented in Section 4, and finally, in Section 5, the main conclusions are drawn.

## 2. Discrete Partial Mutual Information from Mixed Embedding

In this section, we present the measure of discrete partial mutual information from mixed embedding (DPMIME), the parametric significance test, and the resampling significance test used in the DPMIME. We also present performance indices for the causality measure when all the K(K−1) causal effects are estimated for all possible directed pairs of the *K* observed discrete variables.

### 2.1. Iterative Algorithm for the Computation of DPMIME

Let $\{x_{1,t},x_{2,t},\ldots ,x_{K,t}\},t=1,2,\ldots ,n$, be the observations of a stochastic process on *K* discrete random variables $X_{1},X_{2},\ldots ,X_{K}$, typically a multivariate Markov chain. The discrete variables can be nominal or ordinal, and for convenience hereafter, we refer to the data as multivariate symbol sequence.

We are interested in defining a measure for the direct causality from *X* to *Y*, where *X* and *Y* are any of the *K* observed discrete variables. For a sufficiently large number of lags *L*, we formulate the set $W_{t}$ of candidate lag variables that may have information explaining the response *Y* at one time step ahead, $Y_{t+1}$. The set $W_{t}$ has K·L components (’·’ denotes multiplication), $X_{i,t-\tau }$, i=1,…,K, τ=0,…,L−1. The algorithm DPMIME aims to build up progressively the so-called mixed embedding vector, i.e., a subset $w_{t}$ of $W_{t}$ of the most informative lag variables explaining $Y_{t+1}$.

In the first step, the first lag variable to enter $w_{t}$ is the one that maximizes the MI with $Y_{t+1}$,
(1)$$w_{1}=\underset{w\in W_{t}}{arg max}I\left(Y_{t+1};w\right)$$
and $w_{t}=w_{t}^{1}=\left[w_{1}\right]$ (the superscript denotes the iteration, equal to the cardinality of the set). The MI of two variables *X* and *Y* is defined in terms of entropy and probability mass functions (pmfs) as [34]
$$I\left(X;Y\right)=H\left(X\right)+H\left(Y\right)-H\left(X,Y\right)=\sum_{x,y}p\left(x,y\right)\log\frac{p_{X,Y}\left(x,y\right)}{p_{X}\left(x\right)p_{Y}\left(y\right)},$$
where H(X) is the entropy of *X*, the sum is over all values *x* and *y* of *X* and *Y*, $p_{X,Y}\left(x,y\right)$ is the joint pmf of (X,Y), and $p_{X}\left(x\right)$ is the pmf of *X*. The pmfs are assumed to regard the multinomial probability distribution and are estimated by the maximum likelihood estimate, where the probability for each value or pair of values is simply given by the relative frequency of occurrence in the sample (the multivariate symbol sequence). In the subsequent steps, the CMI instead of the MI is used to find the new component to enter $w_{t}$. Suppose that at step *j*, the *j* most relevant lag variables to $Y_{t+1}$ are found forming $w_{t}=w_{t}^{j}$. The next component to be added to $w_{t}^{j}$ is one of the components in $W_{t}\setminus w_{t}^{j}$ (the K·L components except the *j* components already selected) that maximizes the CMI to $Y_{t+1}$, i.e., the mutual information of the candidate *w* and $Y_{t+1}$ conditioned on the components in $w_{t}^{j}$
(2)$$w_{j+1}=\underset{w\in W_{t}\setminus w_{t}^{j}}{arg max}I\left(Y_{t+1};w|w_{t}^{j}\right).$$

The CMI of two variables *X* and *Y* given a third variable *Z* is defined in terms of entropy and pmfs as [34]
$$\begin{array}{cc} I(X;Y|Z) & =-H(X,Y,Z)+H(X,Z)+H(Y,Z)-H(Z)\\ \; & =\sum_{x,y,z}p_{X,Y,Z}\left(x,y,z\right)\log\frac{p_{X,Y,Z}\left(x,y,z\right)p_{Z}\left(z\right)}{p_{X,Z}\left(x,z\right)p_{Y,Z}\left(y,z\right)}. \end{array}$$

At each step, when the lag variable is selected, using (Equation 1) for the first step and (Equation 2) for the subsequent steps, a significance test is run for the MI in (Equation 1) and the CMI in (Equation 2). The parametric and resampling significance tests are presented in detail later in this section. For the step j+1, where $w_{j+1}$ is found in (Equation 2), if the CMI $I(Y_{t+1};w_{j+1}|w_{t}^{j})$ is found statistically significant by the parametric or resampling test, the $w_{t}$ is augmented as $w_{t}=w_{t}^{j+1}=\left[w_{t}^{j},w_{j+1}\right]$. Otherwise, there is no significant lag variable to be added to the mixed embedding vector and the algorithm terminates, giving the mixed embedding vector $w_{t}=w_{t}^{j}$.

The components of the mixed embedding vector $w_{t}$ obtained upon termination of the algorithm are grouped in lag variables of the driving variable *X*, $w_{t}^{X}$, the response variable *Y*, $w_{t}^{Y}$, and all other K−2 variables, $w_{t}^{Z}$, expressed as $w_{t}=\left[w_{t}^{X},w_{t}^{Y},w_{t}^{Z}\right]$. If $w_{t}^{X}$ is empty, i.e., no-lag variable $X_{t-\tau }$ has information to explain $Y_{t+1}$ in view of the other lag variables, there is no direct causality from *X* to *Y*. Otherwise, we quantify the direct causality from *X* to *Y* as the proportion of the information of $Y_{t+1}$ explained by the lag variables of *X*. The measure DPMIME is thus defined as
(3)$$DPMIME_{X\to Y}=\left\{\begin{array}{cc} 0, & \mathrm{if}\;w_{t}^{X}=\emptyset .\\ \frac{I(Y_{t+1};w_{t}^{X}|w_{t}^{Y},w_{t}^{Z})}{I(Y_{t+1};w_{t})}, & \mathrm{otherwise}. \end{array}\right.$$

In the following, we present the resampling test and the parametric test for the significance of the CMI of the response $Y_{t+1}$ and the selected component $w_{j+1}$ given the components already selected in $w_{t}^{j}$, $I(Y_{t+1};w_{j+1}|w_{t}^{j})$.

### 2.2. Randomization Test for the Significance of CMI

First, we do not assume any asymptotic parametric distribution of the estimate of $I(Y_{t+1};w_{j+1}|w_{t}^{j})$ under the null hypothesis $H_{0}:I\left(Y_{t+1};w_{j+1}|w_{t}^{j}\right)=0$. Thus, the empirical distribution of the estimate of $I(Y_{t+1};w_{j+1}|w_{t}^{j})$ is formed by resampling on the initial sample of the variables $Y_{t+1}$, $w_{j+1}$ and $w_{t}^{j}$. For this, we follow the resampling scheme of the so-called time-shifted surrogates for the significance test for correlation or causality [35,36]. The resampling is actually applied only to $w_{j+1}$. To retain both the marginal distribution and intra-dependence (autocorrelation) of $w_{j+1}$ and destroy any inter-dependence to $Y_{t+1}$ and $w_{t}^{j}$, we shift cyclically the symbol sequence of $w_{j+1}$ by a random step *k*[35] (We do not consider here the case of periodic or periodic-like symbol sequences, where this randomization scheme is problematic, as it is likely that the generated surrogate symbol sequence is similar to the original symbol sequence.). Thus, for the original symbol sequence $\left\{w_{j+1,1},w_{j+1,2},\ldots ,w_{j+1,n}\right\}$ of $w_{j+1}$, a randomized (surrogate) symbol sequence for the random step *k* is
$$\left\{w_{j+1,1}^{*},w_{j+1,2}^{*},\ldots ,w_{j+1,n}^{*}\right\}=\left\{w_{j+1,k+1},\ldots ,w_{j+1,n},w_{j+1,1},\ldots ,w_{j+1,k}\right\}.$$

We derive a number *Q* of such randomized symbol sequences and compute for each of them the corresponding estimates of $I(Y_{t+1};w_{j+1}|w_{t}^{j})$ under the $H_{0}$, denoted
$$I\left(Y_{t+1};w_{j+1}^{*1}|w_{t}^{j}\right),I\left(Y_{t+1};w_{j+1}^{*2}|w_{t}^{j}\right),\ldots ,I\left(Y_{t+1};w_{j+1}^{*Q}|w_{t}^{j}\right).$$

These *Q* values form the empirical null distribution of the estimate of $I(Y_{t+1};w_{j+1}|w_{t}^{j})$. The $H_{0}$ is rejected if the estimate of $I(Y_{t+1};w_{j+1}|w_{t}^{j})$ on the original data is at the right tail of the empirical null distribution. To assess this, we use rank ordering, where $r^{0}$ is the rank of the estimate of $I(Y_{t+1};w_{j+1}|w_{t}^{j})$ in the ordered list of the Q+1 values, assuming ascending order. The *p*-value of the one-sided test is $1-\left(r^{0}-0.326\right)/\left(Q+1+0.348\right)$ (using the correction in [37] to avoid extreme values such as p=0 when the original value is last in the ordered list, which is formally not correct). The DPMIME measure in Equation (Equation 3) derived using resampling test of CMI is denoted DPMIMErt.

### 2.3. Parametric Test for the Significance of CMI

Entropy and MI on discrete variables are well-studied quantities and there is rich literature about the statistical properties and distribution of their estimates. For the significance test for the CMI I(X;Y|Z) for three discrete scalar or vector variables *X*, *Y*, and *Z*, the most prominent of the parametric null distribution approximations are worked out in [38], namely the Gaussian and Gamma distributions. For the Gamma null distribution, following the work in [39], it turns out that $\hat{I}\left(X,Y\right)$ follows approximately the Gamma distribution
$$\hat{I}\left(X,Y\right)∼\Gamma \left(\frac{\left(P_{X}-1\right)\left(P_{Y}-1\right)}{2},\frac{1}{n\ln2}\right),$$
where *n* is the sample size and $P_{X}$ is the number of the possible discrete values of *X*. Further, it follows that $\hat{I}\left(X,Y|Z\right)$ is also approximately Gamma distributed
(4)$$\hat{I}\left(X,Y|Z\right)∼\Gamma \left(\frac{P_{Z}}{2}\left(P_{X}-1\right)\left(P_{Y}-1\right),\frac{1}{n\ln2}\right).$$

We use the Gamma distribution to approximate the null distribution of the estimate of $I(Y_{t+1};w_{j+1}|w_{t}^{j})$ for the significance test of CMI, setting $Y_{t+1}$, $w_{j+1}$ and $w_{t}^{j}$ as *X*, *Y*, and *Z*, respectively, in Equation (Equation 4). The parametric significance test is right-sided, as is for the resampling significance test, and the *p*-value is the complementary of the Gamma cumulative density function for the value of the estimate of $I(Y_{t+1};w_{j+1}|w_{t}^{j})$. The DPMIME measure in Equation (Equation 3) derived using the parametric test of CMI is denoted DPMIMEpt. Both tests in the computation of DPMIMErt and DPMIMEpt are performed at the significance level α=0.05.

### 2.4. Statistical Evaluation of Method Accuracy

For a system of *K* variables, there are K(K−1) ordered pairs of variables to estimate causality. In the simulations of known systems, we know the true interactions between the system variables from the system equations. We further assume that the causal effects in each realization of the system match the designed interactions. Though this cannot be established analytically, former simulations have shown that for weak coupling, below the limit of generalized synchronization, the match holds [1]. Thus, we can assess the match of the K(K−1) estimated causal effects to the true causal dependencies using performance indices. Here, we consider the indices of specificity, sensitivity, Matthews correlation coefficient, F-measure, and Hamming distance. All the indices refer to binary entries, i.e., there is causal effect or not, so we do not use the magnitude of DPMIME in (Equation 3), but only if it is positive or not.

The sensitivity is the proportion of the true causal effects (true positives, TPs) correctly identified as such, given as sens = TP/(TP + FN), where FN (false negative) denotes the number of pairs having true causal effects but have gone undetected. The specificity is the proportion of the pairs correctly not being identified as having causal effects (true negatives, TNs), given as spec = TN/(TN + FP), where FP (false positive) denotes the number of pairs found falsely to have causal effects. For the perfect match of estimated and true causality, sensitivity and specificity are one. The Matthews correlation coefficient (MCC) weighs sensitivity and specificity [40]
$$\mathrm{MCC}=\frac{\mathrm{TP}\cdot \mathrm{TN}-\mathrm{FP}\cdot \mathrm{FN}}{\sqrt{\left(\mathrm{TP}+\mathrm{FP})\cdot (\mathrm{TP}+\mathrm{FN})\cdot (\mathrm{TN}+\mathrm{FP})\cdot (\mathrm{TN}+\mathrm{FN}\right)}}$$

MCC ranges from −1 to 1. If MCC = 1, there is perfect identification of the pairs of true and no causality; if MCC = −1, there is total disagreement and pairs of no causality are identified as pairs of causality and vice versa, whereas MCC at the zero level indicates random assignment of pairs to causal and non-causal effects. The F-measure is the harmonic mean of precision and sensitivity. The precision, also called positive predictive value, is the number of detected true causal effects divided by the total number of detected casual effects, F = TP/(TP + FP). The F-measure (FM) ranges from 0 to 1. If FM = 1, there is perfect identification of the pairs of true causality, whereas if FM = 0, no true coupling is detected. The Hamming distance (HD) is the sum of false positives (FPs) and false negatives (FNs). Thus, HD obtains non-negative integer values bounded below by zero (perfect identification) and above by K(K−1) if all pairs are misclassified.

## 3. Simulations

One of the aims of the simulation study is to assess whether and how the DPMIME on discrete-valued time series attains the causality estimation accuracy of PMIME on the respective continuous-valued time series. Therefore, we generate discrete-valued time series on the basis of the causality structure of a continuous-valued time series. The continuous-valued time series is generated by a known dynamical system so that the original causal interactions are given by the system equations. In the simulation study, we consider four different ways to generate discrete-valued time series aiming at having the original causal interactions, as presented below.

(1)*Continuous to Discrete by quantization (Con2Dis)*: The multivariate symbol sequence of a predefined number of symbols *M* is directly derived by quantization of the values of the multivariate continuous-valued time series of *K* observed variables. The range of values of each variable is partitioned to *M* equiprobable intervals and each interval is assigned to one of the *M* symbols.(2)*Realization of estimated sparse Markov Chain (SparseMC)*: The multivariate symbol sequence is generated as a realization of a Markov chain of reduced form estimated on the Con2Dis multivariate symbol sequence (as derived above from the continuous-valued multivariate time series). First, the transition probability matrix of a Markov chain of predefined order *L* is estimated on the Con2Dis multivariate symbol sequence. An entry in this matrix regards the probability of a symbol of the response variable conditioned on the ‘word’ of size K·L of *L* last symbols of all *K* variables. For *M* discrete symbols, the size of the transition probability matrix for one of the *K* response variables is $M^{K\cdot L}\times M$. The causal interactions in the original dynamical system assign zero transition probabilities to words that contain non-existing causal interactions so that the Markov chain has a reduced form as the lag variables are less (or much less for a sparse causality network) than K·L. For example, let us assume the case of K=3, L=2, and M=2 and the true lag causal relationships for the response $X_{1,t+1}$ are $X_{1,t}$, $X_{1,t-1}$, and $X_{2,t}$. The full form of the Markov chain comprises $2^{3\cdot 2}=64$ conditioned probabilities for each of the two symbols of $X_{1,t+1}$, but we estimate only the $2^{3}=8$ probabilities as the lag variables $X_{2,t-1}$, $X_{3,t}$, and $X_{3,t-1}$ are not considered to have any causal effect on $X_{1,t+1}$. Even for a sparse causality network (few true lag causal relationships), the multivariate Markov chain can only be estimated for relatively small values of *K*, *L*, and *M*. Once the sparse transition probability matrix is formed, the generation of a multivariate symbol sequence of length *n* goes as follows. The first *L* symbols for each of the *K* variables are chosen randomly, and they assign to the initial condition. Then, for times t+1, t=L+1,...,n+T, the new symbol of each of the *K* variables is drawn according to the estimated conditioned probabilities. Finally, the first *T* symbols for each variable are assigned to a transient period and are omitted to form the SparseMC multivariate symbol sequence of length *n*.(3)*Realization of estimated mixture transition distribution model (MTD)*: Instead of determining the multivariate Markov chain of reduced form in SparseMC, a specific and operatively more tractable form called mixture transition distribution model (MTD) has been proposed [41]. In essence, instead of determining the transition probability from the word of the causal lag variables to the response variable, the MTD determines the transition probability from each causal lag variable to the response variable. Here, as lag variable, we consider any lag of the driving variable to the response in the true dynamical system, e.g., for the example above for the driving variable $X_{2}$ to the response $X_{1}$, we consider both lags of $X_{2}$ (assuming L=2) and not only the true lag one. In its full form, the MTD assumes that the state probability distribution of the *j*-th variable at time t+1 (response variable) depends on the state probability distribution of all *K* variables at the last *L* times as
$$X_{j,t+1}=\sum_{i=1}^{K}\sum_{l=1}^{L}\lambda_{j,i,l}P_{j,i,l}X_{i,t-l+1},\;i=1,2,\ldots ,K,\;t=L,L+1,\ldots ,$$
where $P_{j,i,l}$ is the transition probability from $X_{i,t-l+1}$ to $X_{j,t+1}$, and $\lambda_{j,i,l}$ is a parameter giving the weight on $X_{i,t-l+1}$ in determining $X_{j,t+1}$, and for j=1,2,…,K, the following holds $\sum_{i=1}^{K}\sum_{l=1}^{L}\lambda_{j,i,l}=1$. We restrict the full form of MTD by dropping from the sum the variables that are non-causal to $X_{j}$, preserving that the remaining $\lambda_{j,i,l}$ sum up to one. Thus, $\lambda_{j,i,l}$ denotes the strength of lag causality from $X_{i,t-l+1}$ to $X_{j,t+1}$. Further, after a simulation study for the optimal tolerance threshold $\lambda_{0}$, we determine $\lambda_{0}=0.01$, and if $\lambda_{j,i,l}<\lambda_{0}$, we set $\lambda_{j,i,l}=0$ to omit terms having small coefficients. In this way, we attempt to retain only significant dependencies of the response on the lag variables. We use the estimated MTD model as the generating process and generate a multivariate symbol sequence. To fit MTD to the Con2Dis multivariate symbol sequence, we use the package markovchain package in R language [42], implementing the fitting of higher-order multivariate Markov chains as described in [43,44].(4)*Realization of estimated multivariate integer-autoregressive system (MINAR)*: Another simplified form of the multivariate Markov chain is given by the multivariate integer-autoregressive systems (MINAR) [19]. Here, we do not estimate MINAR from the Con2Dis multivariate symbol sequence, as done for the sparse multivariate Markov chain (SparseMC) and the MTD process, but define the MINAR of order one, MINAR(1), by setting to zero the coefficients that regard no-lag causality in the original dynamical system. Therefore, the *j*-th variable at time *t* is given as $X_{j,t}=\sum_{i=1}^{K}\alpha_{i,j}\circ X_{i,t-1}+R_{j,t}$ for j=1,…,K, where $\alpha_{i,j}\in \left[0,1\right]$ are the coefficients of MINAR(1) (set to zero if the corresponding driver–response relationship does not exist in the original dynamical system), ∘ denotes the thinning operator (The thinning operator defines that a∘x is the sum of *x* Bernoulli outcomes of probability *a*.), and $R_{j,t}$ is a random variable taking integer values from a given distribution (here, we set the discrete uniform of two symbols). We note that the way the integer-valued sequence is generated does not determine a fixed number of integer values for each of the *K* variables so that the generated multivariate symbol sequence does not have a predefined number *M* of symbols.

The multivariate symbol sequences of all four types are generated under the condition of preserving the coupling structure of the original continuous-valued system. However, only the first type Con2Dis directly preserves the original coupling structure, as the Con2Dis multivariate symbol sequence is directly converted from the continuous-valued realization of the original system. For the other three types, a restricted model is first fitted to the Con2Dis multivariate symbol sequence, which is then used to generate a multivariate symbol sequence. Among the three models, the sparse Markov chain (SparseMC) is best constrained to preserve the original coupling structure. The other two models, the MTD and MINAR, are included in the study as there are known models for discrete-valued time series, adapted here to the given coupling structure. However, the MTD does not preserve the exact lag coupling structure of the original system and the MINAR generates multivariate symbol sequences of varying number of symbols at each realization so that the estimation of the causality structure with the DPMIME on the MTD and MINAR multivariate symbol sequences is not expected to be accurate.

We compute the DPMIME on each multivariate symbol sequence and evaluate the statistical accuracy of the DPMIME to estimate the true variable interactions and subsequently the true coupling network. Further, we compute also the PMIME on the initial continuous-valued time series and examine whether DPMIME can attain the accuracy of PMIME.

### 3.1. The Simulation Setup

In the simulation study, we use as the original dynamical system the coupled Hénon maps [33,45] and consider four settings regarding different connectivity structures for K=5 (here, the *K* variables constitute the *K* subsystems being coupled). We also consider a vector stochastic process as a fifth generating system.

The first system (S1) has an open-chain structure of K=5 coupled Hénon maps, as shown in Figure 1a, defined as
(5)$$\begin{array}{ccc} X_{1,t+1} & = & 1.4-X_{1,t}^{2}+0.3X_{1,t-1}\\ X_{2,t+1} & = & 1.4-{\left(0.5C\left(X_{1,t}+X_{3,t}\right)+\left(1-C\right)X_{2,t}\right)}^{2}+0.3X_{2,t-1}\\ X_{3,t+1} & = & 1.4-{\left(0.5C\left(X_{2,t}+X_{4,t}\right)+\left(1-C\right)X_{3,t}\right)}^{2}+0.3X_{3,t-1}\\ X_{4,t+1} & = & 1.4-{\left(0.5C\left(X_{3,t}+X_{5,t}\right)+\left(1-C\right)X_{4,t}\right)}^{2}+0.3X_{4,t-1}\\ X_{5,t+1} & = & 1.4-X_{5,t}^{2}+0.3X_{5,t-1} \end{array}$$

The first and last variable in the chain of K=5 variables drives its adjacent variable and each of the other variables drive the adjacent variable to its left and right. The coupling strength *C* is set to 0.2 regarding weak coupling.

The second system (S2) has a randomly chosen structure, as shown in Figure 1b, and it is defined as
(6)$$\begin{array}{ccc} X_{1,t+1} & = & 1.4-X_{1,t}\left(\left(1-C\right)X_{1,t}+CX_{3,t}\right)+0.3X_{1,t-1}\\ X_{2,t+1} & = & 1.4-X_{2,t}^{2}+0.3X_{2,t-1}\\ X_{3,t+1} & = & 1.4-X_{3,t}\left(0.5CX_{2,t}+\left(1-C\right)X_{3,t}+0.5CX_{5,t}\right)+0.3X_{3,t-1}\\ X_{4,t+1} & = & 1.4-X_{4,t}\left(0.5CX_{2,t}+0.5CX_{3,t}+\left(1-C\right)X_{4,t}\right)+0.3X_{4,t-1}\\ X_{5,t+1} & = & 1.4-X_{5,t}\left(CX_{4,t}+\left(1-C\right)X_{5,t}\right)+0.3X_{5,t-1} \end{array}$$

The coupling strength *C* is set to 0.5. There is no predefined pattern for the interactions of the variables, other than the number of interactions being six, as for S1.

The other two systems, S3 and S4, also have a randomly chosen structure similar to S1 (see Figure 1c,d). S3 is defined as
(7)$$\begin{array}{ccc} X_{1,t+1} & = & 1.4-X_{1,t}\left(\left(1-C\right)X_{1,t}+CX_{5,t}\right)+0.3X_{1,t-1}\\ X_{2,t+1} & = & 1.4-X_{2,t}^{2}+0.3X_{2,t-1}\\ X_{3,t+1} & = & 1.4-X_{3,t}\left(\left(1-C\right)X_{3,t}+CX_{5,t}\right)+0.3X_{3,t-1}\\ X_{4,t+1} & = & 1.4-X_{4,t}\left(\left(1-C\right)X_{4,t}+CX_{5,t}\right)+0.3X_{4,t-1}\\ X_{5,t+1} & = & 1.4-X_{5,t}\left(\frac{1}{3}CX_{1,t}+\frac{1}{3}CX_{2,t}+\frac{1}{3}CX_{3,t}+\left(1-C\right)X_{5,t}\right)+0.3X_{5,t-1} \end{array}$$
and S4 is defined as
(8)$$\begin{array}{ccc} X_{1,t+1} & = & 1.4-X_{1,t}^{2}+0.3X_{1,t-1}\\ X_{2,t+1} & = & 1.4-X_{2,t}\left(CX_{1,t}+\left(1-C\right)X_{2,t}\right)+0.3X_{2,t-1}\\ X_{3,t+1} & = & 1.4-X_{3,t}\left(CX_{2,t}+\left(1-C\right)X_{3,t}\right)+0.3X_{3,t-1}\\ X_{4,t+1} & = & 1.4-X_{4,t}\left(CX_{3,t}+\left(1-C\right)X_{4,t}\right)+0.3X_{4,t-1}\\ X_{5,t+1} & = & 1.4-X_{5,t}\left(0.5CX_{1,t}+0.5CX_{4,t}+\left(1-C\right)X_{5,t}\right)+0.3X_{5,t-1} \end{array}$$

The coupling strength *C* is set to 0.5 for S3 and 0.4 for S4. System S3 has node 5 as a hub (three in-coming and three out-going connections) and system S4 has a causal chain from node 1, to 2, to 3, to 4.

The fifth system (S5) is a vector autoregressive process on K=5 variables (model 1 in [46]), and it is defined as
(9)$$\begin{array}{ccc} X_{1,t+1} & = & 0.4X_{1,t}-0.5X_{1,t-1}+0.4X_{5,t}+u_{1,t+1}\\ X_{2,t+1} & = & 0.4X_{2,t}-0.3X_{1,t-3}+0.4X_{5,t-1}+u_{2,t+1}\\ X_{3,t+1} & = & 0.5X_{3,t}-0.7X_{3,t-1}-0.3X_{5,t-2}+u_{3,t+1}\\ X_{4,t+1} & = & 0.8X_{4,t-2}+0.4X_{1,t-1}+0.3X_{2,t-1}+u_{4,t+1}\\ X_{5,t+1} & = & 0.7X_{5,t}-0.5X_{5,t-1}-0.4X_{4,t}+u_{5,t+1} \end{array}$$

The terms $u_{j,t+1}$ are white noise with zero mean. The connectivity structure of S5 is shown in Figure 1e.

To derive statistically stable results, we generate 100 realizations for each system and for different time-series lengths *n*. The number of symbols (*M*) for the discretization of the continuous-valued multivariate time series is 2 and 4, respectively. For the discretization, an equiprobable partition is used so that when M=2, all values of the time series larger than the median are set to 0 and the rest to 1, and when M=4, the quartiles of the time series define the four symbols.

### 3.2. An Illustrative Example

The performance of the DPMIME is first illustrated with a specific example, focusing on the first two equations of system S1 and thus considering as a response in the DPMIME (and PMIME) only the first and second variable. We consider only the first type (Con2Dis) for the generation of the multivariate symbol sequence with M=2 symbols and length n=1024. The parametric test for the significance of each component to be added to the mixed embedding vector is used (DPMIMEpt), and the maximum lag is L=5. The PMIME is computed for the same L=5 on the continuous-valued time series (before discretization). Table 1 shows the frequency of occurrence of any of the 25 lag terms in the mixed embedding vector of DPMIMEpt and PMIME for the response $X_{1}$ and $X_{2}$ in 100 Monte Carlo realizations. For the true lag terms, i.e., the terms that occur in the system equations, the frequencies are highlighted.

The variable $X_{1,t+1}$ depends on the variables $X_{1,t}$ and $X_{1,t-1}$, which are selected by both algorithms of DPMIMEpt and PMIME in all realizations (frequency 100). The DPMIMEpt selects also the lag terms $X_{1,t-2}$, $X_{1,t-3}$, and $X_{1,t-4}$ of the response variable, but their inclusion in the mixed embedding vector does not result in any false causal effects. (It turns out that it is hard to find the exact lag components of the driving variable in the case of discrete-valued time series, which questions the use of DPMIME for building the input of a regression model to the response.) No other lag terms are found (the maximum frequency is one).

The second equation of S1 defines the dependence of $X_{2,t+1}$ on $X_{1,t}$, $X_{2,t}$, $X_{2,t-1}$, and $X_{3,t}$. For $X_{2}$ as response, both algorithms do not include the lag term $X_{1,t}$ in the mixed embedding vector (frequency 6 for DPMIMEpt and 1 for PMIME) but include instead $X_{1,t-1}$ (frequency 84 for DPMIMEpt and 99 for PMIME) so that the variable $X_{1}$ is represented in the mixed embedding vector and the correct causal effect from $X_{1}$ to $X_{2}$ is established. The lag terms $X_{2,t}$ and $X_{2,t-1}$ are always present in the mixed embedding vector for both algorithms ($X_{2,t}$ occurs less frequently at 91% for DPMIMEpt) and terms of larger lag of $X_{2}$ occur for DPMIMEpt at a smaller frequency. The representation of $X_{3}$ in the mixed embedding vector is spread over the two first lags for PMIME and to the first four lags for DPMIMEpt so that though the true lag term $X_{3,t}$ is not well identified (frequency 32 and 44 for DPMIMEpt and PMIME, respectively), the causal effect from $X_{3}$ to $X_{2}$ is well established. The variables $X_{4}$ and $X_{5}$ are not represented in the mixed embedding vector, and thus both DPMIMEpt and PMIME correctly find no causal effect from these variables to $X_{2}$.

The example shows that the two algorithms have a similar performance, with DPMIMEpt tending to include more lag terms of the causal variables, but both algorithms do not include lag terms of variables that have no causal effect on the response variable.

### 3.3. System 1

The example above is for the first two variables of S1, and in the following, we compute DPMIMEpt and PMIME for all K=5 response variables of S1 and detect the presence of causal effects by the presence of a lag term (or terms) of the driving variable in the mixed embedding vector for the response variable. The true causal effects as derived by the equations of S1 are $X_{1}\to X_{2},X_{2}\to X_{3},X_{3}\to X_{2},X_{3}\to X_{4},X_{4}\to X_{3}$, and $X_{5}\to X_{4}$. The distribution of the DPMIMEpt and PMIME (in the form of boxplots) and the rate of detection of causal effects (numbers under the boxplots) for all 20 directed variable pairs are shown in Figure 2.

Both measures perfectly define the non-existent causal effects with a percentage of detection less than 3%. The DPMIMEpt detects the true causal effects in high percentages, approaching the perfect identification achieved by PMIME. However, as seen by the size of the boxplots, the DPMIME obtains smaller values than the PMIME. Though both are defined by the same CMI to MI ratio in (Equation 3), this ratio is smaller for the DPMIME.

To quantify the performance of the DPMIMEpt and PMIME at each realization of S1, we calculate the performance indices sens, spec, MCC, FM, and HD on the 20 binary directed connections, where six of them are true. In Table 2, the average indices over the 100 realizations of S1 for n=1024 are shown for both measures.

For the DPMIMEpt using M=2, both sensitivity and specificity are very high, and the overall indices are high as well, e.g., the HD shows that more often none or less often one causal effect out of 20 causal effects is misclassified (average HD is 0.38). Thus, the performance of the DPMIMEpt is close to the almost perfect performance of the PMIME.

Next, we compare the parametric test (PT) and resampling test (RT) for the CMI used in the DPMIME as the criterion to terminate the algorithm building the mixed embedding vector. We consider the four types for generating multivariate symbol sequences (Con2Dis, SparseMC, MTD, and MINAR) and for M=2 and M=4 symbols. The latter does not apply to MINAR as the generated sequences have not a predefined number of symbols (integers). In the comparison, we again use as reference (gold standard) the PMIME, because this is computed directly on the continuous-valued time series, whereas the DPMIMEpt and DPMIMErt are computed on the discrete-valued time series. We also examine the performance of measures for different time-series lengths *n*. Here, we only report results for the performance index MCC in Table 3.

The DPMIMEpt and DPMIMErt fail to define the pairs with causal and non-causal effects when applied to the multivariate symbol sequences generated by the MTD. As already mentioned, the MTD model fails to preserve the causality of the original system and, in turn, the generated discrete-valued sequences do not allow for the estimation of the true causal effects. For example, for M=2 when n=1024, the performance indices sens, spec, MCC, FM, and HD are 0.36, 0.60, −0.03, 0.31, and 9.41, respectively, indicating a very low specificity. The DPMIME using either significance test on the Con2Dis sequences scores similarly in the MCC and at a lower level than the PMIME, converging to the highest level with the increase of *n*. This holds for both M=2 and M=4, but for M=4, the performance of DPMIME is worse than that of PMIME and the difference decreases with *n*, indicating that for a larger number of symbols longer time series are needed. The accuracy of DPMIME on the SparseMC sequences is similar as for the Con2Dis sequences when M=4, but for M=2, the accuracy does not improve with *n* unlike in the case of Con2Dis. The DPMIME performs better on MINAR sequences than on MTD sequences, especially when the resampling significance test is used in DPMIME. In this particular case, the parametric test is not as accurate as the resampling test. The finding that DPMIMEpt and DPMIMErt (except in the case of MINAR) perform similarly has practical importance because we can rely on DPMIMEpt and save computation time, which for long time series and many observed variables, DPMIMErt would be computationally very intensive.

Similar results as for MCC in Table 3 are obtained using the performance index HD, as shown in Figure 3a. The misclassification is larger when the time series gets smaller (from 4096 to 512) for the same number of symbols *M*, and when *M* gets larger (from 2 to 4) for the same *n*. However, the HD is at the same level for all these settings for DPMIMEpt and DPMIMErt, and for both measures, it converges to zero (no misclassification of all 20 variable pairs) for n≥2048, as does PMIME even for small *n*.

### 3.4. System 2

System S2 differs from S1 in that it has a randomly chosen coupling structure. We show the summary results of the performance index MCC in Table 4 and the HD in Figure 3b. The performance of DPMIMEpt and DPMIMErt on the Con2Dis and SparseMC sequences is similar to that on system S1 for the different settings of time-series length *n* and number of symbols *M* (the MTDs are not included in the results due to their poor performance in the previous system). The accuracy in detecting the true causal effects is better for smaller *M* when *n* is small, converging to the highest performance level with *n* and faster for M=4 (MCC = 1 and HD = 0). The highest level is again attained by PMIME even for the smallest time-series length n=512. For the smallest n=512, the performance of DPMIMEpt and DPMIMErt is better for system S2 than for system S1. Another difference to system S1 is that for the largest tested n=4096, the DPMIMEpt and DPMIMErt reach the highest level for M=4 but not for M=2, indicating that once there is enough data, the use of the largest number of symbols allows for a better detection of the causal effects. For system S2, the DPMIMEpt and DPMIMErt on the MINAR sequences give similar MCC scores that do not tend to get higher with *n*, unlike the respective scores for system S1. This lack of improvement with *n* in the causality estimation on MINAR sequences is attributed to the varying number of integers of the generated time series increasing with *n* so that the number of symbols *M* is relatively large compared to the length of time series *n*.

### 3.5. System 3 and System 4

Systems S3 and S4 also have a randomly chosen structure, as with system S2. The summary results of the performance index MCC are shown in Table 5 for S2 and Table 6 for S3.

For the different settings of both S2 and S3, the generation of symbol sequences by Con2Dis and SparseMC, the number of symbols M=2 and M=4, and the time-series lengths *n*, the DPMIMEpt and DPMIMErt always perform similarly and less accurately than the PMIME. There are however differences in the DPMIME in Con2Dis and SparseMC with respect to *M*. As for S1 and S2, for both S3 and S4, the DPMIME on SparseMC symbol sequences tends to perform better for M=4 than for M=2, and this occurs more consistently for a larger *n*. On the other hand, the DPMIME on Con2Dis symbol sequences tends to perform better for M=2 than for M=4, particularly for a smaller *n*. For a larger *n*, for S3, the best performance is observed for Con2Dis and M=2, and for S4, SparseMC and M=4.

### 3.6. System 5

System S5 is a five-dimensional vector autoregressive process of order 4. This system is chosen in order to examine the performance of the causality measures in a linear stochastic system. The summary results of the performance index MCC are presented in Table 7. First, it is worth noting that the PMIME does not reach the highest accuracy level as for the nonlinear deterministic systems S1–S4, but the MCC ranges from 0.76 to 0.78 for the different *n*. The highest accuracy level is attained by the DPMIMEpt for a smaller *n* and the DPMIMEpt for a larger *n* on the Con2Dis symbol sequences when M=4. For M=4, the randomization test tends to outperform the parametric test for a larger *n* and attains the maximum MCC = 1. On the other hand, when M=2, the accuracy of both tests is at the same level and does not improve significantly with the increase of *n* as does for PMIME.

### 3.7. Effect of Observational Noise

In the last part of the simulation study, we investigate the effect of observational noise, restricting to observational noise on the original continuous-valued time series. We consider system S1 and add to each of the five generated time series white Gaussian noise with standard deviation (SD) being a given percentage of the SD of the time series. The Con2Dis approach with M=2 is then applied to derive the symbol sequences of different lengths *n* and the DPMIME is applied using the parametric test (DPMIMEpt) and randomization test (DPMIMErt). In Table 8, the results are presented, including the PMIME measure as well. The type of test does not seem to affect the performance of the DPMIME for all different noise levels. For noise levels up to 10%, the MCC is rather stable and effectively the same as for the noise-free symbol sequences and decreases with a further increase in noise level (20% and 40%) for all different *n*. However, even for the high noise level of 40% when n=4096, the MCC is 0.8 for DPMIMEpt and 0.85 for DPMIMErt and close to the MCC for PMIME at 0.89. Overall, a smooth decrease in the accuracy of the DPMIME is observed with the increase in the level of observational noise, which suggests the appropriateness of DPMIME for real-world symbol sequences.

## 4. Application to Real Data

We consider a real-world application to compare DPMIME to other causality measures. These are the linear direct causality measure called the conditional Granger causality index (CGCI) [47,48], the information-based direct causality measure of partial transfer entropy (PTE) [49], and finally, the original partial mutual information from mixed embedding (PMIME).

The dataset is the Morgan Stanley Capital International (MSCI) market capitalization weighted index of five selected markets in Europe and South America: Greece, Germany, France, UK, and USA. Specifically, we consider two datasets: the first one is in the time period 1 January 2004 to 31 January 2008 and the second one in the period from 3 March 2008 to 30 March 2012. The separation was made with regard to the occurrence of the global financial crisis (GFC), also referred to as the Great Recession, dated from the beginning of year 2008 to year 2013 [50]. The two selected periods are therefore called preGFC and postGFC. The interest here is to study whether and how each of the causality measures detects changes in the connectivity structure in the system of the five markets from preGFC to postGFC. Each dataset comprises *n*=1065 observations, which correspond to daily returns (first differences in the logarithms of the indices). For DPMIME, the data were discretized to two symbols: 1 if the return is positive and 0 otherwise. For consistency, the amount of past information denoted *L* is the same for all causality measures and set to L=2, where for DPMIME and PMIME *L* stands for the maximum lag, for PTE it stands for the embedding dimension, and for CGCI it stands for the order of the (restricted and unrestricted) vector autoregressive (VAR) model.

The causality measures DPMIMEpt, PMIME, CGCI, and PTE are computed for each pair of national markets in the preGFC and postGFC periods. While DPMIME and PMIME assign zero to the non-significant causal relationships, CGCI and PTE require a threshold, here given by the parametric significance test for CGCI and the resampling significance test for PTE (the time-shifted surrogates as for the significance of CMI in DPMIMErt). Then, the causality networks are formed drawing weighted connections with weights being the value of the significant measure, and the networks are shown in Figure 4 for the preGFC and postGFC periods.

All causality measures suggest the USA market has a causal effect on many other markets before and after the GFC. In the DPMIMEpt networks (Figure 4a,e), there is in additional causal effect from UK to Germany in both the preGFC and postGFC periods, while the driving from France to UK in preGFC reverses in the postGFC period. Regarding the latter, the PMIME networks show the opposite, UK to France in preGFC and France to UK in postGFC (Figure 4b,f). The PMIME networks show no causal effect of USA on Greece in both periods, which has no straightforward interpretation. In the postGFC period, the PMIME finds a bidirectional causal relationship for the USA and UK. The CGCI measure gives almost full networks in both periods (Figure 4c,g), failing to reveal any particular connectivity structure in the system of the five national markets. On the other hand, the PTE turns out to be the most conservative measure, giving only the causal effect of the USA to UK, Germany and France (not Greece) in both periods (Figure 4d,h).

The DPMIMErt gave similar results to DPMIMEpt (not shown here). We repeated the same analysis for the DPMIME and L=1, and the results were stable. Overall, the DPMIME estimates reasonable causal relationships, the USA to all four other markets in both periods, whereas the PMIME and PTE exclude Greece, and the UK and France causal relationship changes direction from preGFC to postGFC.

## 5. Discussion

In this study, we propose a Granger causality measure for discrete-valued multivariate time series or multivariate symbol sequences based on partial mutual information from a mixed embedding named DPMIME. The rationale is to build the so-called mixed embedding vector that has as components the lag terms of the observed variables that best explain the response ahead in time. To quantify the causality of a driving variable to a response variable in view of all the observed variables, we first check whether the lag terms of the driving variable are included in the mixed embedding vector. If there are not any, then the measure is zero and there is no causal effect, whereas if there are, then the proportion of the information on the response explained by these lag terms determines the strength of the causal effect from the driving to the response. For the termination of the algorithm building the mixed embedding vector, we develop a parametric test using a Gamma approximation of the asymptotic null distribution of the conditional mutual information, CMI (information of the tested lag term and the response given the other lag terms already included in the mixed embedding vector). This is a main difference to the PMIME, the analogue of the same algorithm already developed by our team for continuous-valued time series. The PMIME employs a resampling significance test as there is no parametric approximation of the null distribution of the CMI for continuous variables. Another main difference to the PMIME is that for discrete-valued time series, we use a different estimate for the information measures of the mutual information, MI, and CMI used in the algorithm, i.e., we use the maximum likelihood estimate for the probabilities of all discrete probability distributions involved in the definition of the MI and CMI, whereas in the PMIME, the nearest neighbor estimate [51] is used for the entropies involved in the definition of the MI and CMI. We develop two versions of the DPMIME, one using the parametric significance test for the termination criterion, denoted DPMIMEpt, and another using the resampling significance test, denoted DPMIMErt, as for the PMIME.

The previous studies of our research team have showed that the PMIME is one of the most appropriate measures to estimate direct causality in multivariate time series and particularly in the setting of high-dimensional time series (many observed variables) [1,27,52]. Therefore, to evaluate the proposed measure DPMIME for the causality of discrete-valued multivariate time series and multivariate symbol sequences, we compare it to the PMIME. For the simulations, dynamical systems of continuous-valued variables were used to generate multivariate time series and compute on them the PMIME. Then, the discrete-valued time series were generated by discretizing the continuous-valued time series, denoted Con2Dis. Moreover, systems for the generation of discrete-valued time series were fitted to the Con2Dis multivariate sequence: the sparse Markov Chain (SparseMC), the mixture transition distribution models (MTD), and the multivariate integer-autoregressive systems (MINAR). The simulations showed that the MTD cannot preserve the original coupling structure, whereas the varying number of integers (assigned to symbols) of the MINAR sequences complicates the use of the DPMIME on these sequences. The SparseMC sequences could preserve sufficiently well the coupling structure in the discrete-valued (Con2Dis) and continuous-valued time series, as the DPMIME could detect the original causality relationships almost as well in the SparseMC sequences as in the Con2Dis sequences. Thus, the main focus in the simulation study was on the performance of the DPMIMEpt and DPMIMErt (for the parametric and resampling significance test) on the Con2Dis multivariate symbol sequences, as compared to the PMIME, where the latter has the role of golden standard as it is computed on the complete available information from the system, i.e., the continuous-valued time series. Further, we assess whether the DPMIMEpt can perform as well as the DPMIMErt, as the DPMIMEpt is much faster to compute and would be preferred in applications.

The simulation systems are coupled Hénon maps of five subsystems, one with an open-chain coupling structure and the other three with a different randomly chosen coupling structure, as well as a vector autoregressive process (VAR) on five variables. The performance indices were computed on binary causality estimates (presence or absence of causal effect) for all pairs of variables (subsystems). The average of the performance indices over 100 realizations for each setting of the time-series length *n* and number of symbols *M* from the discretization were reported. The results on all the simulated systems showed that the DPMIMEpt scores lower than the PMIME, as expected, but converges to the performance level of the PMIME with an increasing *n*, except for the VAR system where the accuracy of the DPMIMEpt in detecting the true causal effects is better when n=1024 and similar to the PMIME when n≥2048. The difference to the PMIME is larger for a small *n* and larger *M* (going from 2 to 4 symbols), which is anticipated as the discretization smooths out information in the time series about the evolution of the underlying system. However, the convergence of the DPMIME to the PMIME for a data size of n≥2048 indicates that the proposed measure can be used in applications with a moderate length of the discrete-valued time series that can have an even high dimension (we tested here for five subsystems).

The finding that the DPMIMEpt and DPMIMErt perform similarly has high practical relevance. The DPMIME is based on multiple computations of the CMI on progressively higher dimensions that are computationally intensive. If we had to rely on the DPMIMErt, the computation time at each iteration of the algorithm would be multiplied with the number of the resampled data used for the resampling significance test. In applications on long sequences of many symbols, the computation time may be prohibitively long using the DPMIMErt with, say, 100 resampling sequences for each test, but it would be approximately 100 times less when using the DPMIMEpt. Thus, the DPMIMEpt is an appropriate measure to estimate the direct causality in many symbol sequences.

The DPMIME was further applied and compared to other causality measures (PMIME, CGCI, and PTE) in one real-world application. We used data from the Morgan Stanley Capital International market capitalization weighted index of five national markets to examine the causality structure of the system of the five markets before and after the start of the global financial crisis. The proposed measure DPMIME detects the crucial role of the US market before and after the start of the global financial crisis without being as conservative as the PTE and without giving full networks as the CGCI.

## Figures and Tables

**Figure 1 entropy-24-01505-f001:**
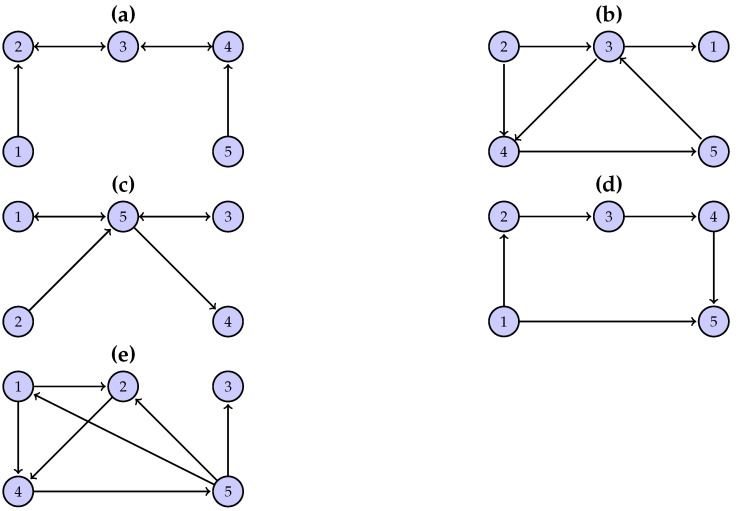
The graphs of the connectivity structure of the simulated systems: (**a**) open-chain structure (S1), (**b**–**d**) randomly chosen structure for (S2)–(S4), respectively, and (**e**) vector autoregressive process (S5).

**Figure 2 entropy-24-01505-f002:**
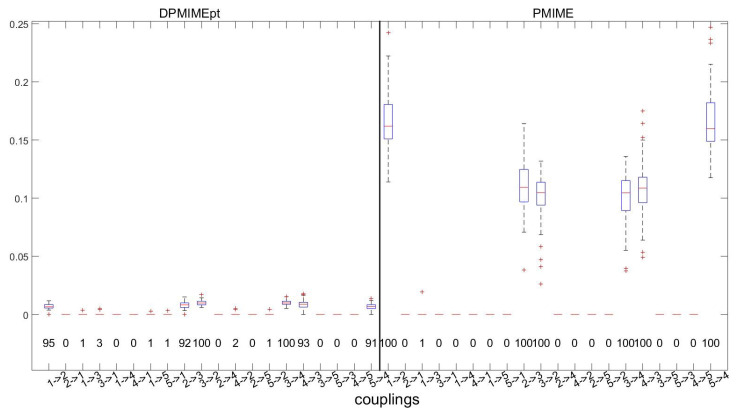
Boxplots of DPMIMEpt (M=2) and PMIME for all variable pairs of S1, for 100 realizations of the system S1, using L=5 and n=1024. At each panel, the number of times the causal effect is detected is displayed below each boxplot.

**Figure 3 entropy-24-01505-f003:**
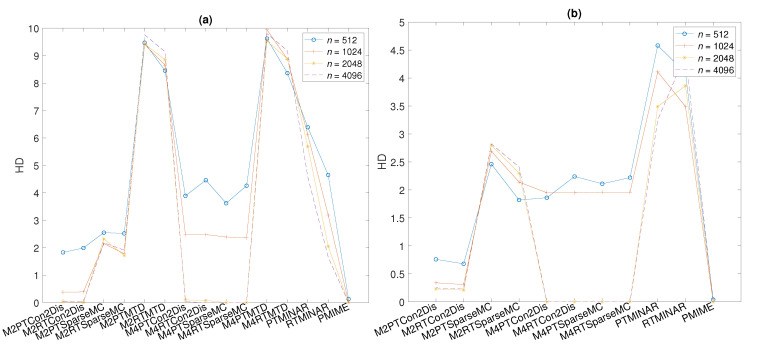
Average HD over 100 realizations of system S1 in (**a**) and S2 in (**b**) for the causality measures DPMIME (L=5) using symbols M=2,4 (denoted with *M* and the number 2 or 4 in the beginning of each word label in the abscissa), the parametric test, and the resampling test (given by PT or RT after the symbol notation of each word label) on multivariate sequences of type Con2Dis, SparseMC, MTD (present only in (**a**)), and MINAR, as well as PMIME (the acronym is at the end of each word label), and for n=512,1024,2048,4096, as given in the legend.

**Figure 4 entropy-24-01505-f004:**
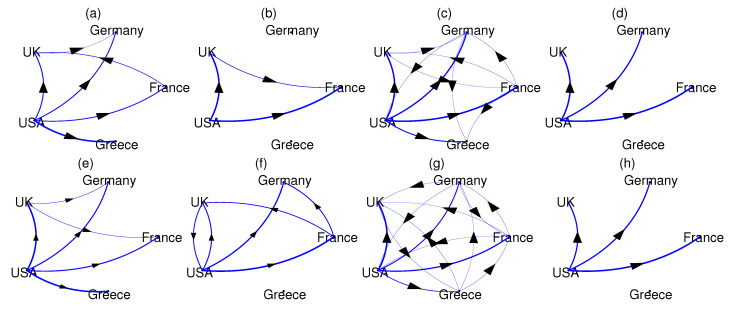
The causality networks of weighted connections for the preGFC period using the measures DPMIMEpt in (**a**), PMIME in (**b**), CGCI in (**c**), and PTE in (**d**), and respectively for the postGFC period in (**e**–**h**).

**Table 1 entropy-24-01505-t001:** Each cell in columns 2–5 has the frequency of occurrence over 100 realizations of the lag variable (first column) in the mixed embedding vector for DPMIMEpt and PMIME, where the response is the first or the second variable of system S1 and for n=1024, L=5, and M=2. The frequencies of the lag variables occurring in the system equations are highlighted.

	$X_{1,t+1}$		$X_{2,t+1}$	
	DPMIMEpt	PMIME	DPMIMEpt	PMIME
$X_{1,t}$	**100**	**100**	**6**	**1**
$X_{1,t-1}$	**100**	**100**	84	99
$X_{1,t-2}$	100	5	3	0
$X_{1,t-3}$	100	1	2	0
$X_{1,t-4}$	92	0	1	0
$X_{2,t}$	0	0	**100**	**100**
$X_{2,t-1}$	0	0	**91**	**100**
$X_{2,t-2}$	0	0	72	5
$X_{2,t-3}$	0	0	8	1
$X_{2,t-4}$	0	0	31	0
$X_{3,t}$	1	0	**32**	**44**
$X_{3,t-1}$	0	0	39	53
$X_{3,t-2}$	0	0	15	6
$X_{3,t-3}$	1	0	10	1
$X_{3,t-4}$	1	0	5	0
$X_{4,t}$	0	0	0	0
$X_{4,t-1}$	0	0	1	0
$X_{4,t-2}$	0	0	0	0
$X_{4,t-3}$	0	0	0	0
$X_{4,t-4}$	0	0	0	0
$X_{5,t}$	0	0	0	0
$X_{5,t-1}$	0	0	0	0
$X_{5,t-2}$	0	0	0	0
$X_{5,t-3}$	0	0	0	0
$X_{5,t-4}$	0	0	0	0

**Table 2 entropy-24-01505-t002:** Average of sensitivity (sens), specificity (spec), MCC, F-measure (FM), and Hamming distance (HD) over 100 realizations of system S1 for the causality measures DPMIMEpt (M=2) and PMIME, using L=5 and n=1024.

	DPMIMEpt	PMIME
sens	0.952	1
spec	0.994	0.999
MCC	0.956	0.999
FM	0.956	0.999
HD	0.380	0.010

**Table 3 entropy-24-01505-t003:** Average MCC over 100 realizations of system S1 for the causality measures DPMIME using L=5, number of symbols M=2,4 (column 1), the parametric test (PT), and the resampling test (RT) (column 2) on multivariate sequences of type Con2Dis, SparseMC, MTD, and MINAR, as well as PMIME (colum 3), and for n=512,1024,2048,4096 (columns 4–7, respectively).

			*n* = 512	*n* = 1024	*n* = 2048	*n* = 4096
*M* = 2	PT	Con2Dis	0.78	0.96	1	1
	RT	Con2Dis	0.76	0.95	1	1
	PT	SparseMC	0.72	0.79	0.79	0.80
	RT	SparseMC	0.71	0.82	0.83	0.82
	PT	MTD	−0.02	−0.03	−0.05	−0.05
	RT	MTD	0.00	−0.01	−0.04	−0.03
*M* = 4	PT	Con2Dis	0.49	0.70	0.99	1
	RT	Con2Dis	0.39	0.70	0.99	1
	PT	SparseMC	0.54	0.72	1	1
	RT	SparseMC	0.43	0.72	1	1
	PT	MTD	0.00	−0.02	−0.03	−0.06
	RT	MTD	0.00	−0.02	−0.02	0.03
	PT	MINAR	0.08	0.12	0.25	0.43
	RT	MINAR	0.41	0.61	0.75	0.81
		PMIME	0.98	1	1	1

**Table 4 entropy-24-01505-t004:** As for Table 3 but for system S2 (MTD not included).

			*n* = 512	*n* = 1024	*n* = 2048	*n* = 4096
*M* = 2	PT	Con2Dis	0.92	0.97	0.98	0.97
	RT	Con2Dis	0.93	0.97	0.98	0.98
	PT	SparseMC	0.76	0.76	0.75	0.75
	RT	SparseMC	0.82	0.80	0.79	0.78
*M* = 4	PT	Con2Dis	0.78	0.77	1	1
	RT	Con2Dis	0.73	0.77	1	1
	PT	SparseMC	0.75	0.77	1	1
	RT	SparseMC	0.74	0.77	1	1
	PT	MINAR	0.29	0.38	0.49	0.54
	RT	MINAR	0.42	0.53	0.49	0.43
		PMIME	1	1	1	1

**Table 5 entropy-24-01505-t005:** As for Table 3 but for system S3 (L=5, MTD and MINAR not included).

			*n* = 512	*n* = 1024	*n* = 2048	*n* = 4096
*M* = 2	PT	Con2Dis	0.8	0.93	0.93	0.95
	RT	Con2Dis	0.78	0.92	0.94	0.95
	PT	SparseMC	0.73	0.75	0.73	0.76
	RT	SparseMC	0.75	0.79	0.77	0.79
*M* = 4	PT	Con2Dis	0.68	0.76	0.81	0.88
	RT	Con2Dis	0.68	0.76	0.81	0.88
	PT	SparseMC	0.65	0.76	0.81	0.88
	RT	SparseMC	0.65	0.76	0.80	0.88
		PMIME	0.98	0.99	1	1

**Table 6 entropy-24-01505-t006:** As for Table 3 but for system S4 (L=5, MTD and MINAR not included).

			*n* = 512	*n* = 1024	*n* = 2048	*n* = 4096
*M* = 2	PT	Con2Dis	0.94	0.98	0.99	0.94
	RT	Con2Dis	0.93	0.98	0.99	0.96
	PT	SparseMC	0.68	0.73	0.70	0.72
	RT	SparseMC	0.74	0.79	0.75	0.75
*M* = 4	PT	Con2Dis	0.82	0.81	0.78	0.70
	RT	Con2Dis	0.86	0.85	0.80	0.72
	PT	SparseMC	0.79	0.87	0.98	1
	RT	SparseMC	0.77	0.87	0.97	1
		PMIME	0.99	0.99	1	

**Table 7 entropy-24-01505-t007:** As for Table 3 but for system S5 (L=8, MTD and MINAR not included).

			*n* = 512	*n* = 1024	*n* = 2048	*n* = 4096
*M* = 2	PT	Con2Dis	0.61	0.71	0.72	0.75
	RT	Con2Dis	0.66	0.74	0.74	0.76
	PT	SparseMC	0.60	0.65	0.68	0.67
	RT	SparseMC	0.66	0.68	0.71	0.70
*M* = 4	PT	Con2Dis	0.81	0.98	0.88	0.86
	RT	Con2Dis	0.75	0.98	1	0.98
	PT	SparseMC	0.82	0.98	0.88	0.86
	RT	SparseMC	0.75	0.98	1	0.98
		PMIME	0.77	0.78	0.78	0.76

**Table 8 entropy-24-01505-t008:** Average MCC over 100 realizations of system S1 for the causality measures DPMIMEpt and DPMIMErt on the Con2Dis symbols sequences (M=2) and PMIME on the original continuous-valued time series (column 2), where noise of different levels is added (column 1), and the time-series length is n=512,1024,2048,4096 (columns 3–6). The added white noise is Gaussian with standard deviation given by the percentage of the standard deviation of the data.

Noise	Measure	*n* = 512	*n* = 1024	*n* = 2048	*n* = 4096
0%	PMIMEpt	0.78	0.96	1	1
	PMIMErt	0.76	0.95	1	1
	PMIME	0.98	1	1	1
5%	PMIMEpt	0.81	0.95	0.98	0.99
	PMIMErt	0.80	0.94	0.98	0.99
	PMIME	0.99	1	1	1
10%	PMIMEpt	0.76	0.95	0.99	0.99
	PMIMErt	0.78	0.95	1	0.99
	PMIME	0.95	0.99	1	1
20%	PMIMEpt	0.72	0.87	0.93	0.93
	PMIMErt	0.71	0.88	0.94	0.94
	PMIME	0.92	0.96	0.97	0.97
40%	PMIMEpt	0.46	0.67	0.76	0.80
	PMIMErt	0.48	0.70	0.80	0.85
	PMIME	0.70	0.84	0.88	0.89

## Data Availability

Not applicable.

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
