# Peer review of "Adaptation of Partial Mutual Information from Mixed Embedding to Discrete-Valued Time Series"

_entropy, 2022, doi:10.3390/e24111505_

Round 1

Reviewer 1 Report

The present manuscript presents a new approach to detect causality among time series by extending the previous PMIME method. Their main contribution is two-fold: on one hand, they present a new variant for discrete instead of continuously-valued time series, which may have a great variety of applications and allows naturally utilizing classical estimators of discrete probabilities in mutual information and conditional variants thereof (as opposed to nearest neighbor based estimates for continuous time series). On the other hand, the new approach allows for deriving a parametric significance test tremendously reducing the computational burden of nonparametric (resampling based) tests necessary for the standard PMIMI method.

I’d like to congratulate the authors to their excellent piece of work and recommend its acceptance for publication after addressing some very minor points in a revised manuscript.

Minor points:

-          L.71: I suppose you mean “discrete partial conditional…”

-          Ll.73-76: Maybe you could already briefly hint here at where the third variables come into play in this formalism. What you outline here reads still like one of the bivariate measures for directional dependence (like standard Granger causality or transfer entropy) that (in their original versions) ignore the effects of any third variables, which is why one may argue that they characterize directional dependence rather than “true” causality.

-          P.3, below the unnumbered equation for I(X;Y): Maybe you could briefly restate what the maximum likelihood estimate is in the present case? In the end, I assume that you consider a multinomial probability model the maximum likelihood parameter estimates of which are exactly given by the empirical frequency distribution of the observed states. Do I understand this correctly?

-          P.4, Section 2.2, line 6 of the paragraph without line numbers: What applying cyclical shifts, I wonder if that may cause any problems with periodic sequences. Could you briefly comment on this?

-          L.121: Can you add a brief note where the “magic numbers” 0.326 and 0.348 of the p-value in the randomization test come from? I understand that the details are given in the reference listed in the same line, but a brief hint on the idea might help clarifying that even resampling tests appear conceptually simple, they are not necessarily.

-          P.5, Section 2.3: Considering I(X;Y|Z), I wonder if the general framework developed here is suitable for only one possible controlling factor Z or also applicable to multiple ones.

-          Same paragraph: Could you leave some brief words on the appropriateness and/or limitations of especially the Gamma approximation, since it is used in the rest of the paper?

-          L.191: How appropriate/typical is the threshold \lambda_0=0.01? Did you do any numerical experiments to find this, or is it “just” experience?

-          L.205: Where do you take mean and variance of R_{j,t} from?

-          Ll. 232, 320, 423: I would not say that S2 has a “random structure”, but a “randomly chosen” structure.

-          In the context of system S2, I would like to understand better if the fact that (unlike S1) this system does not have any bidirectional links contributes to the differential performance of the discussed techniques in both systems. Can the authors give any corresponding reasoning, or at least some educated peculation?

-          Ll.252-254: You consider causal links identified at incorrect links as still correct identifications. To what extent is this justified? I may imagine applications of causal inference tasks where the correct lags are crucial (for example, in climate problems).

-          Section 3.3: What would you consider as an acceptable FPR here, and how could it be controlled in practical applications?

-          Section 4: Should we expect ANY lagged responses across (at least the European) markets in times of high-frequency trading and information practically available in real-time?

-          L.397: Can you give a reference to the used nearest-neighbor estimate? Is it based on Kraskov et a. (2004)?

-          Please check the following references for completeness: 10, 23 (page no./article ID), 26 (article ID), 29 (page no./article ID), 30 (page no./article ID), 31 (article ID), 38 (complete journal name), 49 (article ID)

Technical suggestions:

-          L.11: “the asymptotic distribution”

-          L.14: “suggests that” would read a bit more smoothly

-          L.42: “a multivariate Markov chain”

-          l.102: “one time step ahead”

-          l.194: “in R language”

-          l.199: “SparseMC” (typo)

-          l.203: “driver-response”

-          l.235: “we generate 100 realizations”

-          l.237: “an equiprobable partition”

-          l.244: “…sequence with M=2 symbols and length…”

-          l.251: “on the variables”

-          l.260: “somewhat lower abundance”

-          l.262: “spread over”

-          l.273: “of a lag term”

-          Figure 2, caption: I think that this caption presents a lot of details redundant with the surrounding text.

-          Table 3, caption, l.2: “number of symbols M=2,4”

-          L.292: “and for M=2 and M=4 symbols”

-          L.308: “unlike in the case”

-          L.310: “is not as accurate as the resampling test”

-          L.335: “large compared to”

-          Ll.346-347: “called preGFC…”

-          Ll.363368 and 371-372: I would remove all occurrences of the word “market(s)” in the aforementioned lines to make the text a bit better readable.

-          L.381: “correlate to”

-          L.388: “using a gamma…”

-          L-428: “with increasing n”

-          L.446: “US market”

-          L.447: “being as conservative”

Reviewer 2 Report

The subject of the manuscript is interesting and a good work on the subject could be worth publishing. Unfortunately, various aspects would need improving. I therefore have to recommend a major revision.

Major points.

1 A single numerical example is really not enough to prove the case of the manuscript

2 The system (b) of figure 2 requires much more detailed discussion and investigations. Systems with feedback are well known for being very delicate to handle. Even the concept of causality is not obvious since the influence between the variables becomes circular. It is difficult to believe that the approach is equally valid for this system using the same number of points as the much simpler one of Figure 2 (a). This is particularly important because the practical case discussed in Section 4 contains various feedback loops.

3 The effects of noise and uncertainties in the data should be discussed. I understand that the tests have been performed for ideally exact input data, a condition not realistic n many applications.

Minor points

1 The English needs polishing. Some sentences are difficult to understand. A few examples can be found even in the abstract. The sentences beginning in lines 3 and 14 need revising. In the first sentence of subsection 2.2 overlooked is not the right verb (it is probably meant overviewed).

2 The notation should be made more consistent. The subscript of w sometimes is time and sometimes the number of relevant lag variables.

3 In subsection 2.2 some evidence should be provided to show that the randomization technique used is effective.

Round 2

Author Response

Thank you.